# Assessment of Solar Energy Potential and Its Ecological-Economic Efficiency: Azerbaijan Case

**Mayis G. Gulaliyev [1],\* , Elchin R. Mustafayev [2] and Gulsura Y. Mehdiyeva [3]**

[1]   Methodology and Econometric Issues of State Regulation of Economy Department, Institute of Economics of Azerbaijan National Academy of Sciences, Baku AZ1143, Azerbaijan
[2]   Education for Foreign Students Department, The State Technical University, Baku AZ1073, Azerbaijan; elchin.mustafayev@aztu.edu.az
[3]   Education Department, The Institute of Economics of Azerbaijan National Academy of Sciences, Baku AZ1143, Azerbaijan; gulsura@list.ru
\*   Correspondence: mayis_gulaliyev@yahoo.com

**Abstract:** The paper investigates resource, technical, economic and market potential of solar energy and its ecology and economic efficiency in Azerbaijan. The authors have distinguished six regions in the territory of Azerbaijan with different levels of solar radiation. The resource potential of the regions is calculated by the AutoCad program. As well, technical and economic potential was calculated by special methodology. The authors concluded that Azerbaijan has so much solar energy resource potential and it is possible to replace traditional carbon types of energy with solar energy, even by using the modern technical equipment. However, it is impossible because of economic disadvantages and very low electricity prices. The price of 1 kWh electricity from carbon fuels is several times cheaper than 1 kWh electricity from solar energy. That is why it is difficult to attract investments to develop solar power. There is a necessity to develop a new electricity price policy to stimulate attractiveness of solar power for households and industry.

**Keywords:** Solar energy; carbon energy; resource potential; technical potential; economic potential; market potential

## 1. Introduction

The most important and most promising of renewable energy sources (RES) is solar energy (SE). Other renewable energy sources, including wind, water and bioenergy, are directly related to solar energy. As science and technology are developing, the possibilities of directly generating electricity and heating from solar energy are expanding.

The main arguments of researchers who oppose the prospects and use of solar cells are that (1) solar cells are much more expensive compared to traditional energy sources; (2) the continuous use of solar cells is technologically difficult, since it is impossible to use this energy at night, as well as in the evenings and mornings, when the sun "sets down"; (3) accumulation of electricity from solar cells is also quite expensive; (4) although no waste is released into the environment during the use of solar cells, emissions of harmful gases occur during the preparation of solar cells, and the impact of these gases on global climate change is thousands of times stronger than carbon dioxide; (5) the materials used in solar panels, including tellurium cadmium and selenide, copper, indium and gallium, are quite expensive and have very few natural resources; (6) SE panels should be located on large areas to produce electricity.

Although some oil-rich countries, including Azerbaijan, make some efforts to develop renewable energy sources, the fact that the cost of using traditional energy sources is much lower than the cost of using SE does not contribute to the interest of individual farms and households on investing to use solar cells. However, given the possible impact of using SE on the environment and economic

development in the long term, development of the solar cells should be justified, as fossil fuel natural resources are limited.

The main objective of the study is the assessment of solar energy potential in Azerbaijan and to justify necessity in investing in the development of solar energy systems for electricity and heating production.

The main research questions are: (1) what is the potential of SE in Azerbaijan and can it be used as the country's energy supply is increasing? (2) how appropriate is the use of solar energy in terms of economics and ecology; (3) Is there any prospect of satisfying an essential part of the country's energy demand by electricity from solar cells?

*Literature Review*

The attractiveness of using solar energy lies in the fact that it is several times higher than the volume of global energy consumption, its affordability is relatively simple, and it is environmentally friendly to use. According to the 2000 Report of the UN World Development Energy Assessment, the annual potential of SE is in the range of 1575–49837 exajoules. This is many times greater than the total energy consumed worldwide at the beginning of the 21st century (580.5 eJ) [1] Some researchers believe that SE has even greater potential. For example, V. Smil [2] claims that the SE potential is more than 13,368 exajoules per hour. In 2011, the International Energy Agency (IEA) noted that the development of efficient, inexhaustible and clean technologies for the use of solar energy will have long-term benefits. This will increase the energy security of countries, allowing them to rely on local, inexhaustible energy, largely independent of import sources [3].

The 2030 Solar Thermal Energy Review document states that there are opportunities in construction of buildings that will allow the building's energy needs to be fully satisfied by solar cells. Solar energy will be the main source of energy for buildings in the future [4]. Researchers emphasize the importance of accounting for the use of solar cells in the design and implementation of building construction [5].

In the economic literature, resource potentials are often considered and evaluated as the natural potential of renewable energy [6–8]. For some regions, this potential atlas has been developed [9]. Typically, the resource potential for each renewable energy source is greater than the technical, economic and market potential. It is much more difficult to realize this potential for consumer use, and it is impossible to achieve this fully. For example, the high potential of solar or wind energy does not mean that they can be fully used. This process has technical and economic problems.

It is wrong to assess the economic value of SEs only if costs are less than revenues since this area can stimulate the development of new and knowledge-intensive industries and the creation of jobs in large volumes. However, the use of solar cells requires large investments. Since the construction of a solar power station or wind power plant with a capacity of 1 kW requires an average of 2.2 thousand US dollars. According to İRENA, the normalized cost of energy generated by SPS (Solar Power Stations) in different countries is different and is an average of $ 0.40 per 1 kWh in PV (Photovoltaic) systems [10]. This is still much more expensive than that produced at fuel plants (approximately $0.045). Therefore, the creation of large SPS is less attractive because it requires large investments. The main advantage of solar energy and other types of renewable energy is that it is available to a larger number of subjects. In particular, the availability of SE is even simpler. In most households in villages and towns, in countries that are located in sunny regions, even hundreds of square meters of land can be used to install solar panels. Even the possibility of adapting the installation of PV systems and use in the agricultural sector is studied in the economic literature [11].

The practice of placing such batteries and PVT (Photovoltaic Thermal) systems on the roofs of houses is used in almost all countries. Therefore, the use of PV and PVT systems of small sizes can be simpler and more efficient. For example, in the United States, the cost of 5 kW solar panels is approximately $15,000 [12]. In a country with an average salary of about $3000, the possibility of using PV or PVT systems is wide and not very expensive. However, installing such systems in households in countries that import solar panels and with low wages is an expensive task. On the other hand, the accumulation and storage of energy produced by large solar cells is a serious problem. In most cases, it

is not possible to use a large amount of energy produced by large solar cells without storage. For example, in Germany, most solar energy is produced by small and medium sized solar power plants.

Studies show that the use of SPS is not only less harmful to the environment compared to other types of energy sources but can even have a positive effect by replacing other types of energy sources [10]. The environmental impact of PV, PVT and collector systems on the environment implies the impact of these systems on biodiversity, water use, human health, soil and air quality, transport corridors, land use, terrain, etc. Numerous studies, (e.g., [13–15]) on the environmental impact of SPS have argued that the effects on biodiversity and direct impacts are so weak that they can be ignored.

## 2. Methodology for Assessing the Solar Energy Potential and Assessing the Environmental and Economic Efficiency of Its Use

For example, in the economic literature various methods are proposed to assess the above-mentioned potential levels. The resource potential of solar energy in any country, region or specific area is the amount of energy generated by solar radiation in this geographical area. This volume, in most cases, is measured in kWh. Theoretically, the resource potential of solar energy can be expressed as:

$$RP_{ge} = S * H_r * T_s \tag{1}$$

where, $RP_{ge}$—theoretical natural resource potential of SE in any area, $S$—square of the area, $H_r$—solar radiation intesity (MW/km$^2$), $T_s$—the number of hours of sunshine during the year.

To calculate the technical potential of SE, Lopez et al. proposed methods for a comparative assessment of the technical potential of renewable energy sources [16]. Mwanza et al. used the "massive model" [17]. According to this model, the technical potential of SE is the volume of solar cells, which can be obtained on the condition that losses due to PV technology and other factors, including weather conditions, as well as losses for cooling in any massif are taken into account. In this case, the amount of energy produced by PV technology can be expressed as:

$$E_A = A_{PV} * H_R * \eta_P * \left(1 - \lambda_p\right) * (1 - \lambda_C) \tag{2}$$

where $E_A$—is annual production capacity PV system (kW.h/year), $A_{PV}$—total square of the massive where placed the solar sells (м$^2$), $H_R$—solar radiation valume in this massive within a year (kW.h/м$^2$), $\eta_P$—module efficiency, $\lambda_p$—loss of the module caused by various causes, including surface contamination of the PV battery (as usual 10%), $\lambda_C$—losses due to cooling of PV batteries (as usual 5%).

If module efficiency $\eta_P$ is expressed as a function of nominal efficiency $\eta_r$ which corresponds to $T_r = 25$ Celsius, then

$$\eta_P = \eta_r * \left(1 - \beta * (T_c - T_r)\right) \tag{3}$$

where $\beta$—temperature coefficient of module efficiency, $T_c$—temperature of the module and $T_r$—reasonable temperature.

Income (R) is the amount from the sale of energy generated by SPS. If R > C, then we can conclude that the installation of SPS is useful. One of the most widely used methods for assessing the economic efficiency of SPS is the method of Levelized Cost of Energy (LCOE) [18]. Based on this method, levelized cost of solar energy can be expressed as:

$$LCOE = \frac{\textit{total costs during exploitation}}{\textit{total electricity during exploitation}} = \frac{\sum_{t=1}^{n} \frac{I_t + M_t + F_t}{(1+r)^t}}{\sum_{t=1}^{n} \frac{E_t}{(1+r)^t}} \tag{4}$$

where $I_t$—total amount of investments aimed at the operation of the power plant during the t year of operation; $M_t$—expenses for the operation and maintenance of the power plant during the t year of operation; $F_t$—the cost of fuel used in the power plant during the t year of operation; $r$–discount rate; $E_t$—the amount of electricity generated by the power plant during the t year of operation; $n$–expected life of the power plant. For solar power plants $n = 20$–40 year, $F_t = 0$. It should be noted that LCOE is

changing dramatically from country to country, and a decrease in its cost means an increase in the economic efficiency of SPS.

The repairs or cleaning required during the operation of the panels also occur without harming the environment. A 2018 U.S. Energy Association report states that solar panels and collectors are environmentally friendly. The only drawback is the disposal of batteries and collectors that are unusable. Some researchers argue that the temperature in the territories where the SPS is located drops slightly [19].

The environmental effectiveness of SPS is also a measurable indicator. In its simplest form, this indicator can be expressed as a decrease in the total carbon emissions if energy production at thermal electric stations is equal to production at SPS in the "*n*" year:

$$Carbon = \sum_{t=1}^{n} E_t * N_t * \alpha_t * \beta_t \tag{5}$$

where, *Carbon*—decreasing carbon emissions from the use of SPS in the "n" year, $E_t$—the volume of electric energy produced in the SPS in the "t" year, $N_t$—number of sunny days, $\alpha_t$—the amount of fuel used during the production of one unit of electric energy at Thermal Power Plants (TPPs) in the "t" year, $\beta_t$—the amount of carbon emissions into the environment from the combustion of one unit of fuel in the "t" year.

The use of SPS means a certain fuel economy. The use of this amount of fuel for other purposes (for example, for export), can lead to additional income for the country. The amount of such income (Revenues) for the "n" year can be calculated as:

$$Revenues = \sum_{t=1}^{n} E_t * \lambda_t * price_t \tag{6}$$

where $E_t$—the amount of electricity produced in the SPS in the "t" year, $\lambda_t$—the amount of fuel needed to produce one unit of electricity in a TPP in the "t" year, $price_t$—the cost of one unit of fuel in the world market in the "t" year. Thus, the use of solar cells can bring both economic and environmental benefits.

*2.1. Assessment of the Resource Potential of Solar Energy in Azerbaijan*

The most important step to expand the use of the economic and market potential of solar cells in Azerbaijan was the launch of the AZGÜNTEX Co solar panel plant in the country in the second half of 2012. The first line in the plant was commissioned in 2012, and the second in 2014, which allowed it to produce 100,000 panels with a power of 200–250 W annually. Thus, the total potential of the AZGÜNTEX solar panel plant is 50 MW of power per year and allows the production of 200,000 panels of 200–250 W each year.

The air temperature in the direction of the mountains in Azerbaijan drops to 4–5 °C at an altitude of 2000 m, and at an altitude of 3000 m it is 1–2 °C. In the coldest month of the year (January), the average monthly temperature on the plains and in foothill areas does not fall below 0 °C. Even in the Absheron peninsula, in coastal areas and islands south of it, it is 3–4 °C. As the height increases, the temperature decreases and at 2000 m above sea level it is (−5 °C)–(−6 °C) (−7 °C in the Nakhchivan Autonomous Republic), and at about 3000 m it is (−12 °C)–(−3 °C). To calculate the resource potential of SE in the economic regions of Azerbaijan, we will use the formula (1).

*2.2. Ecological-Economic Efficiency of Using Solar Energy in Azerbaijan*

The efficiency of using solar cells in Azerbaijan can be estimated on the basis of calculating its economic and subsequent market potential. There is no doubt that there are broad prospects for renewable energy sources since there are problems with the supply of fuel to TPPs that use oil, gas and coal, and also because these stations seriously affect climate change. In particular, the results of calculations of the potential of SE for economic regions also show that Azerbaijan has great potential to meet a significant part of its energy needs through SPS.

Calculations show that the resource potential of solar cells in Azerbaijan is more than $18 * 10^7$ million MWh per year (Table 1). For comparison, we note that the volume of energy consumed in Azerbaijan in the field of economic activity in 2017 amounted to $11 * 10^7$ MWh, and the amount of energy consumed by households was $3 * 10^7$ MWh. In other words, the annual resource potential of solar cells more than a million times exceeds the amount of energy consumed in Azerbaijan. The calculated technical potential of SE throughout the country is also much larger than the volume of consumption.

Calculations, taking into account areas with the possibility of using solar energy from a technical perspective and the efficiency of modern PV technologies, show that the technical potential of solar cells in Azerbaijan is at least 20 times less than its resource potential and can be estimated at $10^7$ million MWh. In such calculations, cultivated areas ($20.5 * 10^9$ m$^2$), forest areas ($10.4 * 10^9$m$^2$), reservoirs ($0.4 * 10^9$ m$^2$) and areas of high mountain regions are deducted from the total area. There is no doubt that the resources and technical potential of solar cells in Azerbaijan can fully satisfy its energy needs.

The main problem with the use of SE is its economic feasibility and the necessary amount of investment in this area. The technical potential in Azerbaijan allows for achieving SE not only at the level of consumption, but even more than 100,000 times higher than this level. However, for the installation and maintenance of PV or PVT panels, in order to obtain the required amount of solar cells consumption, more than \$20 billion is required.

Since the average cost of 6 kW panels on the world market is about \$3600 without construction, transportation and other costs [20], such panels are capable of generating 1.7 MWh of electricity per year. It is impossible to invest such an amount in the construction of the SPS in the current conditions. It should also be kept in mind that the same amount will need to be invested in 25–30 years because the service life of modern PV and PVT is an average of 25 years.

Currently, Azerbaijan's wealth of oil and gas is decreasing interest in the creation of SPS and large investments. In 2017, energy products in Azerbaijan's energy balance, which is equivalent to oil with a total production of 64.9 million, accounted for 87.9% of primary energy products, 8.6% of oil refining products, and 3.5% of thermal and electric energy. Of the primary energy products, 69.8% were crude oil (including gas condensate), 29.7% natural gas, and 0.5% energy products produced from renewable energy sources. Azerbaijan not only fully satisfies domestic energy demand, but also exports large volumes of oil and gas. In 2017, exports from the country amounted to 43.3 million tons of oil equivalent, of which 78.0 percent accounted for crude oil, 19.1 percent for natural gas, 2.6 percent for oil products and 0.3 percent for electricity. Households accounted for 41.3% of final energy consumption, 13.4% for industry and construction, 31.2% for transport and 14.1% for other sectors of the economy.

The total amount of pollutants emitted into the atmosphere in 2017 amounted to $36.87 * 10^6$ kg, according to the data in the section "Supply with electricity, gas, vapors and air conditioning" of State Statistical Committee of Azerbaijan Republic [21]. The volume of electricity produced by the country's leading TPP that year amounted to $20445.4 * 10^6$ kWh. In other words, in the production of each kilowatt-hour of electricity, $1.8 * 10^{-3}$ kg = 1.8 g of carbon emissions were generated. In the same year, at the Azerbaijan Thermal Power Plant, a leading enterprise in the energy industry, $336.18 * 10^{-3}$ kg of oil equivalent was burned for the production of 1 kWh of electricity. Thus, let's take the average service life of modern PV and PVT panels to be 25 years (n = 25), the number of sunny hours in Azerbaijan is on average 2000 hours per year, and the average carbon emissions generated in the production of 1 kWh of electricity are $\beta_t = 1.9 * 10^{-3}$ kg, and suppose that the volume of electricity demand does not change during operation, then the positive impact of E = 6 kWh of PV panels on the environment (i.e., decreasing carbon emissions) will be:

$$C = n * E * NS * \beta = 25 * 6 * 2000 * 1.8 * 10^{-3} \approx 540 \text{ kg}$$

This means 21.6 kg of carbon emissions per year.

**Table 1.** Distribution of solar energy resource potential by territorial groups in the Azerbaijan.

| | I Group | | II Group | | III Group | | IV Group | | V Group | | VI Group | |
|---|---|---|---|---|---|---|---|---|---|---|---|---|
| | Indicator | Thousand km² | Indicator | Thousand km² | Indicator | Thousand km² | Indicator | Thousand km² | Indicator | Thousand km² | Indicator | Thousand km² |
| Number of sunlight hours (hour) | 1800–2000 | 8.358 | 2000–2200 | 26.072 | 2200–2400 | 45.676 | 2400–2600 | 2.790 | 2600–2800 | 2.442 | 2800 and more | 0.844 |
| Number of non-sunny days (day) | 10–30 | 1.852 | 30–50 | 14.971 | 50–70 | 63.615 | 70–90 | 13.816 | 90–110 | 2.018 | - | - |
| Direct solar radiation incident on a horizontal surface in cloudless weather (W/m²) | 556–580 | 8.747 | 580–604 | 18.365 | 604–628 | 16.345 | 628–652 | 30.971 | 652–676 | 6.497 | 676–700 | 5.346 |
| Direct solar radiation incident on a horizontal surface and partly cloudy (W/m²) | 194–218 | 1.714 | 218–242 | 34.902 | 242–266 | 29.342 | 266–290 | 14.752 | 290–314 | 2.956 | 314–338 | 2.578 |
| Scattered solar radiation in cloudless weather (W/m²) | 250–274 | 2.314 | 274–298 | 21.710 | 298–322 | 34.410 | 322–346 | 11.420 | 346–370 | 9.788 | 370–394 | 2.364 |
| Scattered solar radiation in partly cloudy (W/m²) | 161–185 | 2.110 | 185–209 | 9.253 | 209–233 | 8.649 | 233–257 | 13.573 | 257–281 | 28.872 | 281–305 | 20.659 |
| General solar radiation in cloudless weather (W/m²) | 806–854 | 6.238 | 854–902 | 21.916 | 902–950 | 37.128 | 950–998 | 11.303 | 998–1046 | 4.653 | 1046–1094 | 1.239 |
| General solar radiation in partly cloudy (W/m²) | 355–403 | 2.904 | 403–451 | 36.674 | 451–499 | 24.490 | 499–547 | 4.896 | 547–595 | 15.766 | 595–643 | 1.541 |

Note: Calculated by authors by using AutoCAD program based on National Atlas [22].

If we take into account that the total electricity demand in the country is approximately 20,000 GWh, the number of PV panels with a capacity of 6 kW needed to provide such an amount of electricity is $20*10^{12}$: (6*2000) = 1.7 million. At the current cost of PV panels, this means more than \$20 billion in investment. Although in practice the installation of PV panels of such a volume goes beyond the economic opportunities of Azerbaijan, theoretically this means a reduction in carbon emissions by 36.9 thousand tons per year.

More sunny days in a large part of Azerbaijan and the accessibility for most households to have the financial resources necessary for installing solar panels expands opportunities for reverse centralization in energy supply. A comparative analysis of the use of electricity by household income in Azerbaijan gives reason to say that for most households, PV panels with a capacity of 1–3 kW can fully satisfy their electricity needs (Table 2). The number of households with an annual consumption of less than 3000 kWh makes up more than 60% of the total number of households in Azerbaijan. Due to the large number of households in Baku and the limited territorial ability of individual households to place PV panels, it is advisable to implement such processes in the vicinity of Baku and regions. Azerbaijan is quite unsuitable for agriculture, is far from water sources, and is even far from low mountainous regions to ensure the production of electricity in large volumes for industrial purposes.

As can be seen from Table 2, household use of PV panels to meet their energy needs through solar cells can be economically justified only under certain conditions. The cost of buying and installing a PV panel with a power of 230–250 W produced in Azerbaijan will cost about \$500 with imported inverters from Chine and Turkey. Obviously, households with incomes up to 250 manat per capita may not be interested in investing this amount.

On the other hand, if the installation of such panels is due to large loans for most households, its attractiveness will become even less. With the current bank interest rate on loans (at best 12%) of 4000–10,000 manat (\$2352–\$5882) and its partial repayment in 25 years of operation, it will be several times higher than the current electricity bill. Households with electricity costs from 88 to 293 manat per year using solar panels and trying to meet their annual electricity needs with 4–9 PV panels will pay between 54–144 manat plus a monthly discount for 25 years for PV panels. Taking into account the fact that at present such households pay monthly between 7 and 24 manat for electricity, the economic nature of households' lack of interest in PV panel projects will become obvious.

If the installation of PV panels will be provided in the form of an interest-free loan for 25 years, then monthly payments will be significantly reduced and will range from 13 to 36 manat. Even then, the monthly payment almost doubles the current interval of 7–24 manat. However, calculations show that the cost of 1 kWh of electricity can be 12 kopecks (1 kopeck= 0.588 sent) if the price of electricity in Azerbaijan corresponds to real market prices.

In this case, the monthly payments of households for consumed electricity generated from fuel come close to payments for PV panel projects. If bank loans or private investment projects for PV panels are designed for a period of less than 25 years, then the use of solar cells can be economically disadvantageous since it is much higher than current electricity prices.

It should be noted that the competitive price for electricity in Azerbaijan is more than 12 kopecks per 1 kWh. Even at such prices, the use of solar panels is more expensive than energy derived from fuel. In the case of such prices, the payback period for installing PV panels is about 25 years. During the same period, the life of PV panels expires. Calculations show that the justification of investment projects related to the use of PV panels in Azerbaijan is possible only if the price of electricity exceeds 40 kopecks. This is an unacceptable price for Azerbaijani households in the current economic and socio-economic situation.

## 2.3. Market Potential

Three important issues in economic regions for calculating the market potential of solar cells in Azerbaijan should be taken into account: (1) the competitiveness of solar cells with other energy sources, especially with thermal power plants running on fuel; (2) the attitude of investors towards

the development of this sector; and (3) the level of regulation. Given that the electricity sector in Azerbaijan and the public sector as a whole are in the state's natural monopoly, electricity prices are set by the tariff council, so all three of the above issues should be determined by the state outside the needs of the free market.

On the other hand, it should be kept in mind that electricity prices in Azerbaijan are much lower than prices on the free market [23], and that the government needs to provide additional compensation to producers to maintain a high level of social security for the population. Consequently, the calculation of the market potential of SE can only matter if the country's electricity sector is based on free market relations.

Despite this, it can be assumed that the state's natural monopoly in the electric power industry is being replaced by liberal market relations. In this case, the calculations show that prices can rise sharply compared to monopoly prices. Therefore, the prices for electricity generated by SPSs can compete with the prices of electricity generated by TPSs [24].

Let's use the method of Levelized Cost of Energy-LCOE, i.e., equation (4), to calculate the average cost of 1 kWh of electricity produced by SPS in Azerbaijan. For solar power plants, we can take an average of n = 25. Assuming that at the initial stage, importing of PV panels with capacity 6 kW will require $12,000 and for installation and other accessories will require about $2000. Assume that the discount rate is zero, and maintenance of the panels is not costly. Even so, for electricity generated in 2000 hours annually in a 25-year period, we get

$$LCOE = \frac{15000\$}{6 * 2000 \, KWh} = 1.25 \frac{\$}{KWh}$$

This is $1250 per 1 MWh and 30 times more than the price of 1 MWh of the energy generated by TPSs, that costs $41 for the fully regulated electricity sector of Azerbaijan.

If the panels costs are covered by state investments, such a project can be completely profitable for 25 years as a result of transferring the remaining part of the household's electricity to the grid after satisfying an average consumption of 1263 kWh. However, the household's consumption is not the same for every hour or minute, so it is necessary to be connected to a centralized electricity grid. One refrigerator, one iron or one vacuum cleaner, connected at the same time, cannot be provided with electricity produced by a 1 kW panel system. Therefore, the profitability of PV panels can be ensured by using the network at a certain time and issuing additional energy to the network at another time.

The situation is compounded by the fact that to ensure profitability, investments are made through bank loans because interest rates in this case should be such that repayment of the loan at a discount does not exceed annual household electricity bills. That is, the installation of PV panels with a loan at 12% of annual rates cannot be effective for households. Depending on the income, interest rates can be adjusted in such a way and the annual repayment rate is determined so that the cost of PV panels paid off within 25 years of operation.

**Table 2.** Estimated costs of installing PV panels and consumed electricity for households with different incomes (2017).

| | Total Consumption (KWh) | Capacity of a PV Panel (W) | A Mount of Hours of Sunshine (hours) | Technical Potential of PV Panels (KWh) | Requared Number of PV Panels | Total Costs (manat) | Annual Costs for Electricity (manat) | Average Discount | Additional Annual Costs with Discount (manat) | Average Monthly Fee (with discount) (manat) | Average Monthly Fee (without discount) (manat) |
|---|---|---|---|---|---|---|---|---|---|---|---|
| | | | | $EPV_t$ | $NPV_t$ | $PVCost_t$ | $ECost_t$ | | $D_t$ | | |
| 115.1–120.0 | 1263 | 250 | 2000 | 500 | 4 | 4054.9 | 88.4 | 12 | 486.6 | 54.1 | 13.5 |
| 120.1–125.0 | 1316.7 | 250 | 2000 | 500 | 4 | 4178.4 | 92.2 | 12 | 501.4 | 55.7 | 13.9 |
| 125.1–130.0 | 1370.4 | 250 | 2000 | 500 | 4 | 4301.9 | 95.9 | 12 | 516.2 | 57.4 | 14.3 |
| 130.1–140.0 | 1451 | 250 | 2000 | 500 | 4 | 4487.3 | 101.6 | 12 | 538.5 | 59.8 | 15.0 |
| 140.1–150.0 | 1558.5 | 250 | 2000 | 500 | 4 | 4734.6 | 109.1 | 12 | 568.1 | 63.1 | 15.8 |
| 150.1–160.0 | 1665.9 | 250 | 2000 | 500 | 4 | 4981.6 | 116.6 | 12 | 597.8 | 66.4 | 16.6 |
| 160.1–180.0 | 1827.1 | 250 | 2000 | 500 | 5 | 5352.3 | 127.9 | 12 | 642.3 | 71.4 | 17.8 |
| 180.1–200.0 | 2042 | 250 | 2000 | 500 | 5 | 5846.6 | 142.9 | 12 | 701.6 | 78.0 | 19.5 |
| 200.1 and more | 2417.5 | 250 | 2000 | 500 | 6 | 6710.3 | 169.2 | 12 | 805.2 | 89.5 | 22.4 |
| 250 and more | 2954.7 | 250 | 2000 | 500 | 7 | 7945.8 | 206.8 | 12 | 953.5 | 105.9 | 26.5 |
| 300 and more | 4184.6 | 250 | 2000 | 500 | 9 | 10,774.6 | 292.9 | 12 | 1292.9 | 143.7 | 35.9 |

Note: calculated by authors.

## 3. Conclusions

Nevertheless, it is important to consider several factors for assessing and improving the economic efficiency of solar cells. There is a need to increase the production capacities of plants producing PV panels several times. It also means creating more jobs. In a plant with a potential capacity of 30 MW, the number of employees will be at least 130, therefore a multiple increase in potential will increase the number of employees almost to the same extent. In such cases, even economies of scale can have an effect and lead to lower marginal costs. Initially, the installation and connection of PV panels to the power grid should be implemented on an investment project implemented by state-owned companies because investing in such a volume by households or enterprises is very high compared to current electricity prices and is economically disadvantageous. Private companies and households should be supported and motivated by a special program for the use of SE. One of the main motivation mechanisms is the installation of PV panels at the expense of state investments and the acceptance of payments for used electricity. The responsibility of households or business structures in this project can only be providing space for PV panels (for example, the possibility of placing roofs on private houses) and ensuring periodic maintenance of panels. The loss of electricity can be minimized, since the unused part of the electricity produced on such panels can be returned to the public electric grid. The amount of payments on the basis of the design for the installation of solar panels should correspond to the current prices of used electricity and should not exceed this amount. This event will also be aimed at motivating households and enterprises to install SE panels. Consumers will not be interested in using SE in cases where prices vary significantly. Regular adjustment of prices for the cost of electricity received from fuel or their adjustment based on market prices will make consumers' choice between electricity from different sources pointless. On the contrary, the constant conduct of activities on solving environmental problems and advocacy outreach can expand the possibilities of using PV panels. Given these conditions, we can determine the minimum cost-effective prices for investment projects for the installation of solar panels.

**Author Contributions:** Conceptualization, G.G.M.; methodology, G.G.M.; software, M.Y.G.; validation, M.Y.G. ; formal analysis, M.R.E.; investigation, M.R.E.; resources, M.R.E.; data curation, M.Y.G.; writing—original draft preparation, G.G.M.; writing—review and editing, G.G.M.; visualization, M.Y.G.; All authors have read and agreed to the published version of the manuscript.

**Funding:** This research received no external funding.

**Conflicts of Interest:** The authors declare no conflict of interest.

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
