# Peer review of "Assessment of Solar Energy Potential and Its Ecological-Economic Efficiency: Azerbaijan Case"

_sustainability, doi:10.3390/su12031116_

Round 1

Reviewer 1 Report

This article authored by Mayis et al. reports an assessment of solar energy potential in Azerbaijan. The authors found that solar energy has a great potential to place traditional fossil fuels in Azerbaijan. They further concluded that the realization of such potential required new electricity price policy to stimulate the implementation of solar energy. This manuscript is well prepared and is recommended for publication after revisions.

[1] Page 1, line 39. What is “honey” used for fabricating solar panels? Is this a typo or do the authors mean something else?

[2] Page 2, lines 80-81. The terms SPS and PVT needs to be spelled out when they first appeared.

[3] Page 2, lines 80-81. The average price of $0.131 per kWh for a PV system is a little bit overestimated. The current estimation is around $0.40 per kWh, according to https://www.eia.gov/outlooks/aeo/pdf/electricity_generation.pdf, and https://www.irena.org/documentdownloads/publications/re_technologies_cost_analysis-solar_pv.pdf.

[4] Page 3, Line 123. What is the term “efficiance”? Is this a typo?

[5] Page 6, Line 203 and page 8 Line 296. “The average cost of 6 kW panels on the world market is about $12,000” overestimated the price of solar panels. Please see http://pvinsights.com/RetailerPrice.php. A 6 kW solar panels cost about $3,600 instead of $12,000. I would suggest the authors double-check their results based on the corrected cost of solar panels.

Author Response

[1] Page 1, line 39. What is “honey” used for fabricating solar panels? Is this a typo or do the authors mean something else?

Response 1: "honey" is incorrect translation. Please accept as "copper"

[2] Page 2, lines 80-81. The terms SPS and PVT needs to be spelled out when they first appeared.

Response 2: These are changed as SPS (Solar Power Stations) and PVT ( Photovoltaic thermal)

[3] Page 2, lines 80-81. The average price of $0.131 per kWh for a PV system is a little bit overestimated. The current estimation is around $0.40 per kWh, according to https://www.eia.gov/outlooks/aeo/pdf/electricity_generation.pdf, and https://www.irena.org/documentdownloads/publications/re_technologies_cost_analysis-solar_pv.pdf.

Responce 3: These sentence is changed as :"According to İRENA, the normalized cost of energy generated by SPS (Solar Power Stations) in different countries is different and is an average of $ 0.40 per 1 kWh in PV (Photovoltaic) systems  (IRENA, 2018)

[4] Page 3, Line 123. What is the term “efficiance”? Is this a typo?

Response 4: efficiance is incorrect. Please accept as "efficiency" 

[5] Page 6, Line 203 and page 8 Line 296. “The average cost of 6 kW panels on the world market is about $12,000” overestimated the price of solar panels. Please see http://pvinsights.com/RetailerPrice.php. A 6 kW solar panels cost about $3,600 instead of $12,000. I would suggest the authors double-check their results based on the corrected cost of solar panels.

Response 4: There was changed as "Since the average cost of 6 kW panels on the world market is about $3,600 without construction costs (PVinsights, 2020)". 

Reviewer 2 Report

In this manuscript, the authors investigate the resource, technical, economic and market potential of solar energy and its ecology and economic efficiency in Azerbaijan. The authors have distinguished six regions in the territory of Azerbaijan with the different levels of solar radiation. The resource potential of the regions is calculated by the Auto-Cad program. As well as there were calculated technical and economic potential by special methodology. There was a necessity to develop new electricity price policies to stimulate attractiveness of solar power for households and industries. The manuscript is well organized and contains interesting findings. 

The main concerns are listed below:

The abstract should be specific and scientific information. The introduction establishes the context of the work, but in the present state, it does not provide sufficient justification for this study. It should be rewritten to expound the research significance of the present work.  How to reduce the emission of carbon in the production of solar energy. The authors should provide a clear schematic representation of the production of solar energy in Azerbaijan. The conclusion part should be specific. In the current state, there are more typographical errors and the language should be improved. Therefore, the authors are advised to recheck the whole manuscript for improving the language and structure carefully.

Author Response

The abstract should be specific and scientific information. The introduction establishes the context of the work, but in the present state, it does not provide sufficient justification for this study. It should be rewritten to expound the research significance of the present work.  How to reduce the emission of carbon in the production of solar energy. The authors should provide a clear schematic representation of the production of solar energy in Azerbaijan. The conclusion part should be specific. In the current state, there are more typographical errors and the language should be improved. Therefore, the authors are advised to recheck the whole manuscript for improving the language and structure carefully.

Response: We edited the introduction, including there was changed objective and research questions.